# Digitalization of Distribution Transformer Failure Probability Using Weibull Approach towards Digital Transformation of Power Distribution Systems

**A. M. Sakura R. H. Attanayake [1,2,\*] and R. M. Chandima Ratnayake [1,\*]**

[1] Department of Mechanical and Structural Engineering and Materials Science, University of Stavanger, P.O. Box 8600 Forus, N-4036 Stavanger, Norway
[2] Ceylon Electricity Board, Distribution Division 04, Dehiwala 010350, Sri Lanka
[\*] Correspondence: sakura.attanayake@ceb.lk (A.M.S.R.H.A.); chandima.ratnayake@uis.no (R.M.C.R.)

**Abstract:** Digitalization of the failure-probability modeling of crucial components in power-distribution systems is important for improving risk and reliability analysis for system-maintenance and asset-management practices. This paper aims to implement a Python programming-based Weibull approach for digitalization of distribution-transformer (DT) failures, considering a regional section of DTs in Sri Lanka as a case study. A comprehensive analysis for DT-failure data for six years has been utilized to derive a Weibull distribution analysis for DTs. The interpretation of the resulting beta and alpha parameters of the Weibull analysis for different categories of DTs in the selected region is also presented. The resulting data can be uploaded to computerized maintenance-management systems (CMMS), to adopt conclusions or resolutions reached by the asset and maintenance managers. Ultimately, failure-probability modeling is beneficial for decision-making processes for higher management aiming for the digital transformation of power-distribution systems.

**Keywords:** CMMS; digitalization; distribution transformers; failure probability; risk and reliability; Weibull approach

## 1. Introduction

Power-distribution systems are involved in the transport of electricity to consumers using hierarchical voltage levels, i.e., from high to low [1–5]. Initially, information about power-distribution networks that had been built decades ago was stored in documents and detailed maps [3]. With the dawn of the era of computers, that information is progressively being digitalized [3]. The required amount of information for assets is large, and the criticality of failures makes operation less complex in low-voltage systems than in high- and medium-voltage systems [3]. Hence, efforts have been made to digitalize the high- and medium-voltage systems [3]. Currently, technologies are evolving to digitalize low-voltage systems [3]. However, power-distribution systems are still far from fully digitally transformed systems with real-time operation [3,6,7]. In the scientific-research context, mathematical-programming models are being developed for the digital transformation of power-distribution systems [3]. This manuscript aims at developing such a mathematical-programming model, which can be considered a stepping stone towards the digital transformation of power-distribution systems.

In addition, some electricity-supply contracts with consumers include clauses that oblige electricity-distribution utilities to pay compensation to consumers in the event of a disruption in the supply of electricity [8]. This depends on the jurisdiction and the specific contract between the electricity-distribution utility and consumers [8]. In general, such clauses, including penalty charges for the electricity-distribution utility, are included in the contract to protect the consumer from such scenarios, and there can be laws or regulations in place in a specific jurisdiction that require or allow for such compensation

to be paid in certain circumstances to the electricity consumers [8]. In addition to the suppliers' losses resulting from failure of equipment, those penalty charges and consumer compensations can be a huge burden to the electricity-distribution utilities [8]. Therefore, in order to manage the circumstances, it is essential that they carry out optimal operation and maintenance of equipment with modern technologies. Hence, the reliability of components in electricity-distribution utilities is very important, as failure of those components is associated with very high costs [9,10].

The hierarchical structure of power-distribution systems is composed of several distinct levels of components, such as distribution transformers (DTs), feeders, switches, fuses, and consumer-side equipment [1]. DTs are crucial equipment in power-distribution utilities, as they constitute high investments and transform primary distribution voltages to lower voltages suitable for consumer equipment [2,11,12].

DT failures cause serious problems to distribution-network management, resulting in comparatively huge expenditure for the repair or replacement of other components in the distribution system [2,9,10]. The reliability analysis of DTs involves the assessment, avoidance, and governance of failures, along with science, statistics, and risk analysis [9,10,13]. Through the use of failure analysis for DTs, the lifespan can be predicted by managing the life cycle and the uncertainties involved with the failure [9,10,13].

There is no zero rate of failure for any equipment [14,15]. Reliability analysis requires time-to-failure data to reveal the patterns of failure occurrences, balancing the reduction in the cost of failures with performance enhancement for digital transformation from the pillars of digitization and digitalization [13,14,16]. Predictions of reliability allow the likelihood of the characteristics of failure rates to be evaluated [13,15]. It is necessary to digitize the wear curves based on the failure data to implement reliability curves of components in power-distribution systems [13]. In addition, it has been observed that extensive study of real DT-failure data analysis using reliability wear curves for the digitalization of power-distribution systems is not addressed in the literature. This manuscript represents the first step in this endeavor.

Based on these needs, this paper has the following objectives: identify DT-category failures that are required for reliability analysis, execute pre-processing and database implementation, determine the probability distributions that can be used, implement Python programming-based reliability-analysis approaches using Weibull distributions and survival functions for different categories of DT failures, and interpret the results for data-driven decisions for the digitalization of the DT-failure analysis.

## 2. Theoretical Background

This section describes the intentions and interpretations connected with the establishment of the approach suggested by this paper. Although DTs can be repaired and reused, in this context, they are considered non-repairable components, due to the fact that, when a failure occurs, they are required to be replaced immediately with a new DT.

### 2.1. Failure-Rate Model

The reliability of a component is the probability of its successful operation until a specific time [17]. It is necessary to obtain historical failure data about failure modes to determine the component reliability [17]. Therefore, it is necessary to study how failures occur over time for life cycle analysis [17].

In addition, every piece of equipment will fail during its operational life [18]. Accordingly, the failure rate of components is given by the following formula [18]:

$$\text{failure rate, } \lambda = \frac{number\ of\ failures}{total\ operating\ time\ of\ units} \tag{1}$$

The relationship between failure, fault, error, and time to failure is illustrated in Figure 1 for continuous and discrete scenarios [19].

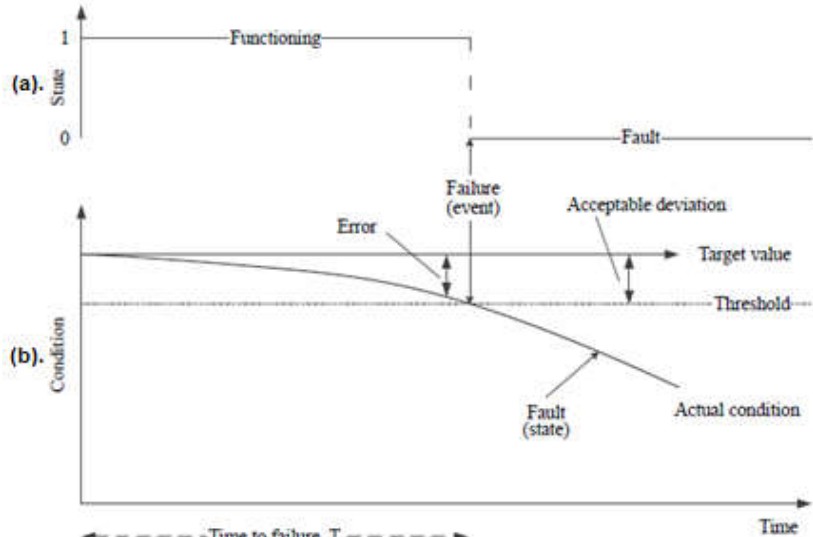

**Figure 1.** Graphical relationship between failure and fault for (**a**) for discrete variables and (**b**) continuous variables [19].

### 2.2. Reliability Function

The reliability functions, $R(t)$ for continuous variables and $R(k)$ for discrete variables, of a component are given by the following formula [20,21]:

$$R(t) = P(T > t) = 1 - F(t), \quad R(k) = P(T > k) = 1 - F(k),$$
$$R(t) = \int_t^\infty f(t)dt, \quad\quad\quad R(k) = \sum_{i=k+1}^\infty f(ki) \quad\quad (2)$$

Here, $F(t)$ indicates the cumulative distribution function (CDF), and $f(t)$ is the probability density function (PDF) and $Pr(T > t)$ yield the probability of a non-failure over time [20,21].

### 2.3. Bathtub Curve

The operational life of a piece of equipment has three important characterizable phases, namely, the infant-mortality phase, useful-life phase, and wear-out phase [18]. A visual representation of the failure rate of a component over time is given by a bathtub curve, as illustrated in Figure 2.

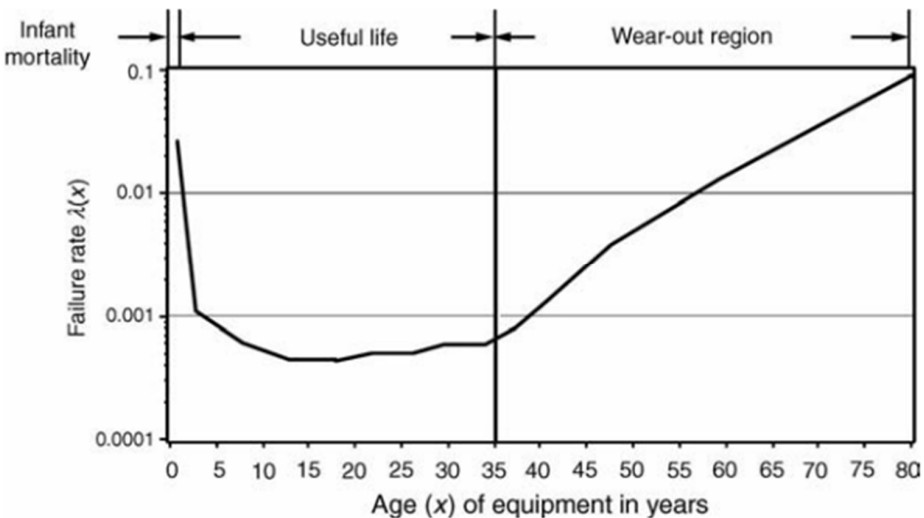

**Figure 2.** Bathtub-curve failure rate versus time [18].

There are three phases of the bathtub curve, which is so-named due to its characteristic bathtub shape [21]. The first phase represents early failures due to poor design and installation. Second-level failures occur in the useful-life period and at random times [18]. The last phase is wear-out failures, where the failure rate rises rapidly due to equipment deterioration [18].

*2.4. Parametric Lifetime Models*

Different parametric lifetime models exist because the different types of data and failure mechanisms occurring over time can be better described graphically by different distributions [17]. For failure data, a particular distribution is assumed in parametric lifetime models for non-repairable components [19].

In parametric lifetime models, the alpha, beta, and gamma parameters are used to characterize the shape of a PDF. Generally, the alpha parameter is the scale parameter that controls the spread of the distribution, while the beta parameter is the shape parameter, which controls the skewness of the distribution [17]. The gamma parameter is an additional shape parameter that allows more flexibility in fitting the data [17]. The selection of a specific parametric model depends on the availability of data, the convenience of analysis of the model parameters, and the capability of the model to accurately describe the lifetime data [17]. Brief details of some of the most common distributions are given below.

2.4.1. Exponential Distribution

This distribution represents the useful-life period in the bathtub curve [19]. The probability-density function (PDF), cumulative-distribution function (CDF), failure-rate function (FRF), and the mean time to failure (MTTF) are given by the following formulae, respectively [19]. Here, the time to failure $t$ is considered a random variable, and $\lambda$ is the scale parameter [19].

$$f(t; \lambda) = \begin{cases} 0 & for\ t < 0 \\ \lambda e^{-\lambda t} & for\ t \geq 0 \end{cases} \tag{3}$$

$$f(t; \lambda) = \begin{cases} 0 & for\ t < 0 \\ 1 - e^{-\lambda t} & for\ t \geq 0 \end{cases} \tag{4}$$

$$h(t; \lambda) = \frac{f(t)}{R(t)} = \frac{\lambda e^{-\lambda t}}{e^{-\lambda t}} = \lambda \tag{5}$$

$$MTTF = \int_0^\infty R(t)\,dt = \frac{1}{\lambda} \tag{6}$$

2.4.2. Weibull Distribution

This distribution represents the whole curve in the bathtub curve [21], with shape parameter $\alpha$ and scale parameter $\beta$, where $\alpha > 0$ and $\beta > 0$ are defined for the probability-density function $f(t)$, cumulative-distribution function $F(t)$, and failure-rate function $h(t)$, as provided below [17,19].

$$f(t; \alpha; \beta) = \begin{cases} 0 & for\ t < 0 \\ \frac{\alpha}{\beta}\left(\frac{t}{\beta}\right)^{\alpha-1} exp[-\left(\frac{t}{\beta}\right)^{\alpha}] & for\ t \geq 0 \end{cases} \tag{7}$$

$$f(t; \alpha; \beta) = \begin{cases} 0 & for\ t < 0 \\ 1 - exp[-\left(\frac{t}{\beta}\right)^{\alpha}] & for\ t \geq 0 \end{cases} \tag{8}$$

$$h(t; \alpha; \beta) = \frac{\alpha}{\beta}\left(\frac{t}{\beta}\right)^{\alpha-1} for\ t > 0 \tag{9}$$

### 2.4.3. Other Parametric Distributions

There are several other parametric distributions, such as normal distribution, logistic distribution, lognormal distribution, gamma distribution, beta distribution, loglogistic distribution, Gumbel distribution, etc. [13,17,19]. For the exponential, Weibull, gamma, lognormal and loglogistic distributions, there is a location-shifting parametrization using the gamma $\gamma$ parameter [13].

Some distributions are preferred to others, depending on the scientific context of the advantages and disadvantages [19]. Exponential distribution is a memoryless distribution and is used to model the lifetime of electronic and electrical equipment that fails randomly [17]. It considers that the past history of the component has no effect on its future lifetime [17]. The normal distribution is capable of describing some dynamic component failures or failures that occur in a specific period of time, including some deviations [17]. The graphical plot of the logistic distribution is analogous to the normal distribution [17]. The failures that happen at the beginning of the life cycle in projects, startups, installations, or operations are mostly represented by the lognormal distribution [17]. It can model both random and wear-and-tear failure mechanisms. The loglogistic distribution and lognormal distributions have the same shape [17]. The Weibull distribution is frequently used to model the lifetime of components that fail due to wear and tear, and it depends on parameters of exponential, lognormal, or normal distributions [17]. The Weibull distribution is flexible and can be used to model increasing and decreasing failure rates [17]. When the data are right-skewed and the failure rate decreases over time, then the gamma distribution can be used for modeling [17]. The component failures that occur at the end-of-life cycle of pipelines, vessels, and towers, etc. can be represented by the Gumbel distribution [17].

### 2.4.4. Goodness of Fit

There are several methods for measuring the goodness of fit of a Weibull model [22]. The methods used are log likelihood, Akaike information criteria (AIC), Bayesian information criterion (BIC) and Anderson-Darling goodness of fit statistic (AD). Log likelihood is the logarithm of the likelihood function, which is the probability of observing the data, given the parameters. Log likelihood is used to compare different models and to determine the best-fitting parameters for a given model [22].

The AIC is a model-selection criterion that balances the goodness of fit of a model with the complexity of the model. It is defined as $-2$ times the log likelihood of the model plus twice the number of parameters in the model [22]. The goal is to find the model with the smallest AIC. Lower AIC values indicate a better model [22].

The BIC is similar to the AIC but also takes into account the number of observations in the data set. It is defined as $-2$ times the log likelihood of the model plus the number of parameters in the model times the natural logarithm of the number of observations. The BIC also aims to find the model with the smallest value [22].

The AD is a statistical test that compares the sample data to a specific probability distribution, such as the Weibull distribution [22]. It measures the difference between the CDF of the sample data and the CDF of the hypothesized distribution, providing a measure of the goodness of fit of the hypothesized distribution to the sample data [22].

### 2.5. Reliability Indices

The ability to maintain continuous service of an electricity-supply system is an indication of the reliability [23]. The following reliability indices are incorporated to measure system performance [19,23]. The failure rate is denoted by $\lambda i$, the number of customers is given by $Ni$, the annual unavailability is denoted by $Ui$, and the average load connected to load point $i$ is given by $La(i)$ [19,23]:

System Average Interruption Frequency Index (SAIFI):

$$\text{SAIFI} = \frac{total\ number\ of\ customer\ interruptions}{total\ number\ of\ customers\ served} = \frac{\sum \lambda i Ni}{\sum Ni} \tag{10}$$

System Average Interruption Duration Index (SAIDI):

$$\text{SAIDI} = \frac{sum\ of\ customer\ interruption\ durations}{total\ number\ of\ customers} = \frac{\sum U_i N_i}{\sum N_i} \qquad (11)$$

Consumer Average Interruption Duration Index (CAIDI):

$$\text{CAIDI} = \frac{sum\ of\ customer\ interuption\ durations}{total\ number\ of\ customer\ interruptions} = \frac{\sum U_i N_i}{\sum \lambda_i N_i} \qquad (12)$$

Energy Not Supplied Index (ENS):

$$\text{ENS} = total\ energy\ not\ supplied\ by\ the\ system = \sum L_{a(i)} U_i \qquad (13)$$

### 2.6. Digitization, Digitalization, and Digital Transformation

The establishment of the digital representation of non-digital things and attributes is known as digitization and is considered a foundational approach for a digital-transformation pyramid [16]. The digital-transformation pyramid is depicted in Figure 3. In this paper, the DT-failure data were converted from paper documents to electronic data for digitization.

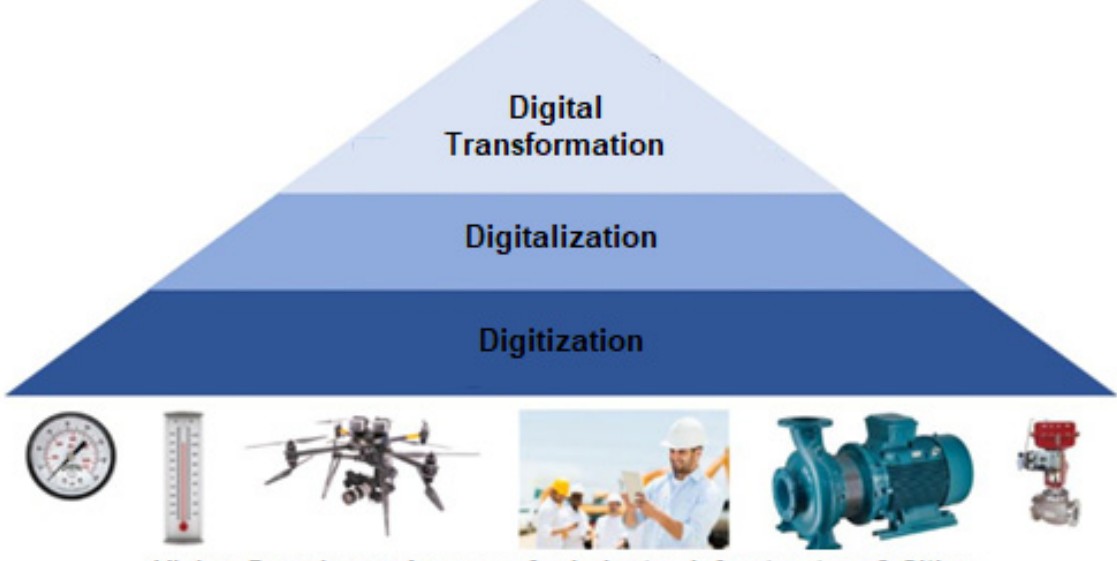

**Figure 3.** Digital-transformation pyramid [16].

Digitalization enables or improves processes, by supporting digital technologies and digitized data, and converts processes from human-driven to software-driven [16]. Hence, digitalization hypothesizes digitization, improves current processes without transforming, and increases productivity and efficiency with a reduction of costs [16]. Digital transformation involves changing business processes by using digitalization technologies [16].

### 3. Case-Study Methodology and Development

#### 3.1. Use of Case-Study Methodology

Usually, case-study methodology is used for research questions with "how" and "why" questions [24]. In this case study, the answers to "how" and "why" aim at analyzing DT-failure data, which were considered with an inductive approach.

Inductive research is a method of reasoning in which the researcher begins with a specific observation and uses it to form a general conclusion. From this approach, the researcher can move from specific observations to a more general theory [24].

"How" and "why" research questions are types of questions that researchers use to guide their research [24]. "How" questions are used to understand the processes or mechanisms that underlie a phenomenon and can be used in inductive research to generate hypotheses or theories based on the observed data [24]. They are frequently used to investigate cause-and-effect relationships and often begin with words such as "how," "what," or "which" [24]. "Why" questions are used to understand the reasons or causes behind a phenomenon [24]. They frequently begin with the word "why" and are used to investigate the underlying reasons for a particular event or outcome [24]. For the inductive approach, the researcher may begin with observations of a certain phenomenon, using them to generate possible explanations or causes for that phenomenon [24]. In both cases, the researcher uses the observations as the basis for generating hypotheses or explanations, which can then be tested through further research [24]. Accordingly, this approach involves providing general conclusions based on specific observations of "how" and "why" DT failures occur.

### 3.2. Weibull Cumulative-Distribution Function and Reliability Function

In this paper, two distribution functions, namely, the Weibull cumulative-distribution function (CDF) and the corresponding reliability function, have been used for the digitalization of transformer failures [22,25,26]. DT-failure data have been recorded in databases from 1 September 2016 to 31 October 2022 from the Western Province South II branch of Ceylon Electricity Board, and those data have been used for the analysis. The Western Province South II branch is having similar geographical and weather conditions in all over the region. Therefore, this region has been considered as acceptable for the reliability analysis of the DT-failure data.

The following categories of transformers were recorded as showing failures for the past six years. The apparent power categories of DTs with failure data for the DT voltages 11 kV/0.4 kV are provided in Table 1.

**Table 1.** The apparent power categories of 11 kV/0.4 kV DTs with failure data.

| Apparent power (kVA) category | 100 | 160 | 250 | 400 | 630 |
|---|---|---|---|---|---|

The apparent power categories of DTs with failure data for the DT voltages 33 kV/0.4 kV are provided in Table 2.

**Table 2.** The apparent power categories of 33 kV/0.4 kV DTs with failure data.

| Apparent power (kVA) category | 100 | 160 | 250 | 400 | 630 | 800 | 1000 |
|---|---|---|---|---|---|---|---|

The failure data of the above categories of DTs were analyzed using Python program-based Weibull analysis. Weibull CDF analysis was selected due to its versatility and modeling capability for a variety of life cycle behaviors of crucial components [13].

The alpha and beta parameters for Weibull distributions can be calculated using maximum likelihood estimation (MLE) or the method of moments, in which sample moments (e.g., mean, standard deviation, etc.) are matched with the theoretical moments of the distribution [19]. The "fitters" library in the Python programming language provides easy-to-use interfaces for fitting Weibull distributions, as well as estimates for the alpha and beta parameters, including other useful features such as goodness-of-fit tests, plots, and confidence intervals with optimization algorithms [19]. Hence, the "fitter" library in the Python programming language -has been used to plot the Weibull CDFs and survival functions in this manuscript. The resulting Weibull CDF plots for DT-failure analysis are provided in Figures 4–9.

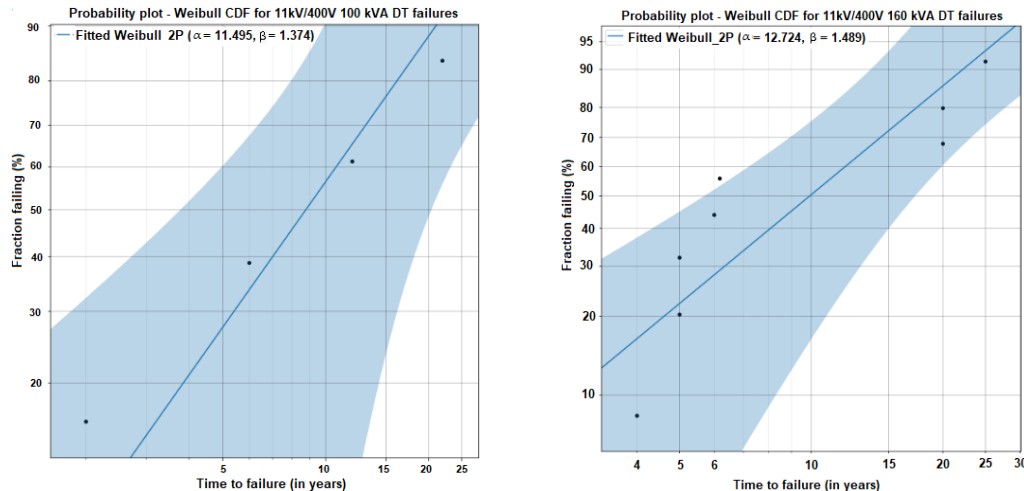

**Figure 4.** Weibull CDF for 11 kV/0.4 kV–100 kVA and 160 kVA DT failures.

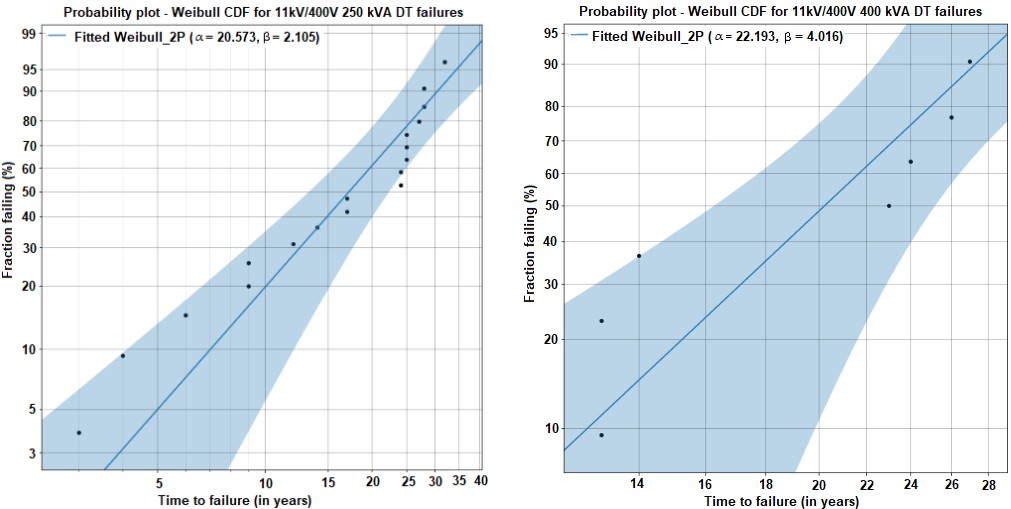

**Figure 5.** Weibull CDF for 11 kV/0.4 kV–250 kVA and 400 kVA DT failures.

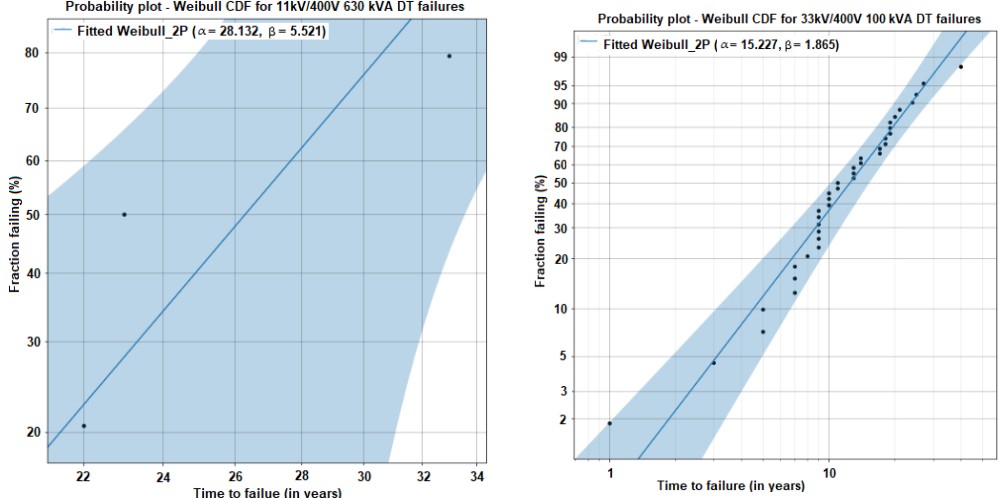

**Figure 6.** Weibull CDF for 11 kV/0.4 kV–630 kVA and 33 kV/0.4 kV–100 kVA DT failures.

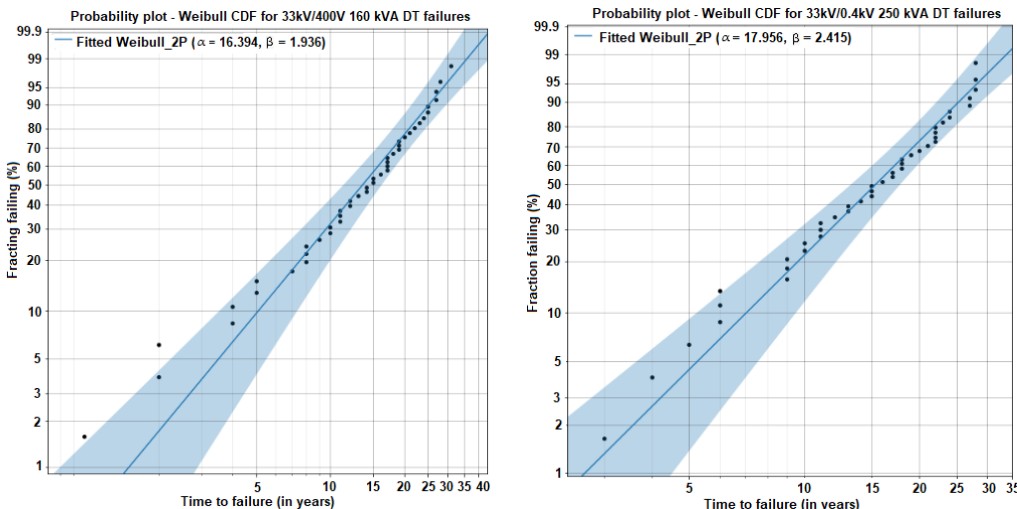

**Figure 7.** Weibull CDF for 33 kV/0.4 kV–160 kVA and 250 kVA DT failures.

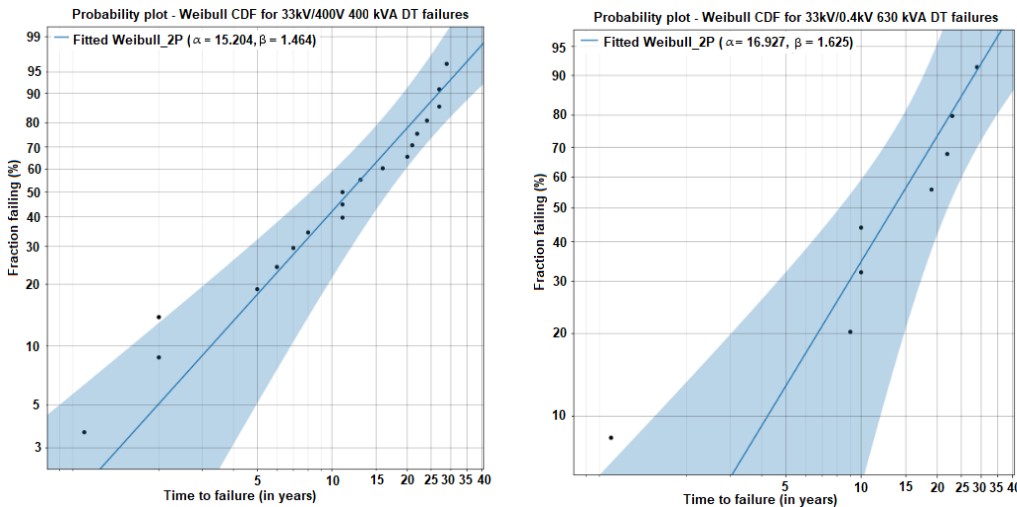

**Figure 8.** Weibull CDF for 33 kV/0.4 kV–400 kVA and 630 kVA DT failures.

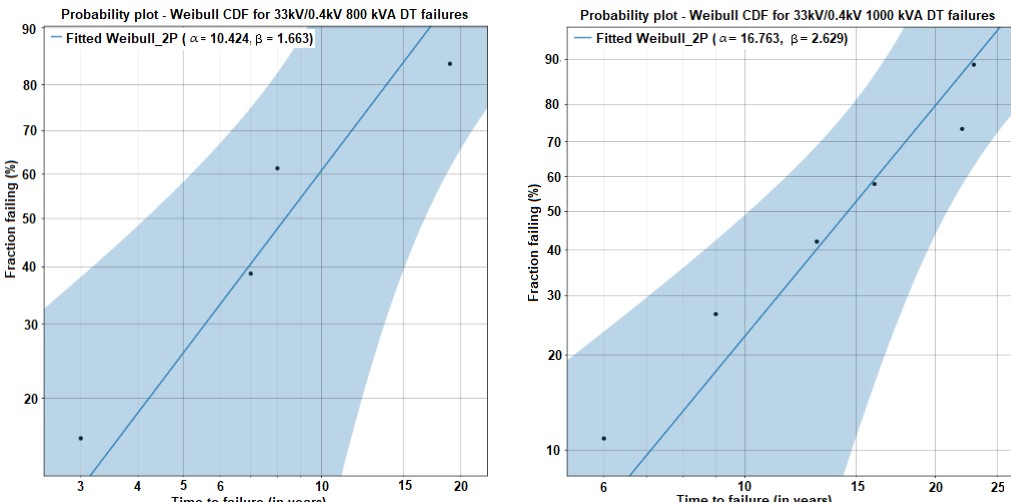

**Figure 9.** Weibull CDF for 33 kV/0.4 kV–800 kVA and 1000 kVA DT failures.

The corresponding survival (reliability) functions were developed using the Python programming language for the different categories of transformer-failure data, as shown in Figures 10–15. To investigate the performance comparison of the different Weibull models for each category of distribution transformer, a single Weibull CDF has been used. The resulting plot is provided in Figure 16.

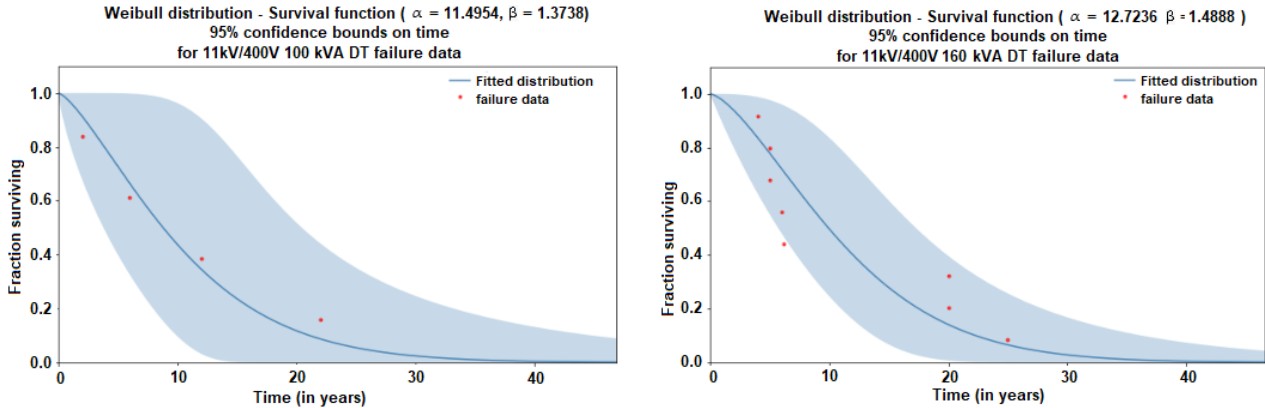

**Figure 10.** Weibull survival functions for 11 kV/0.4 V–100 kVA, 160 kVA DT failures.

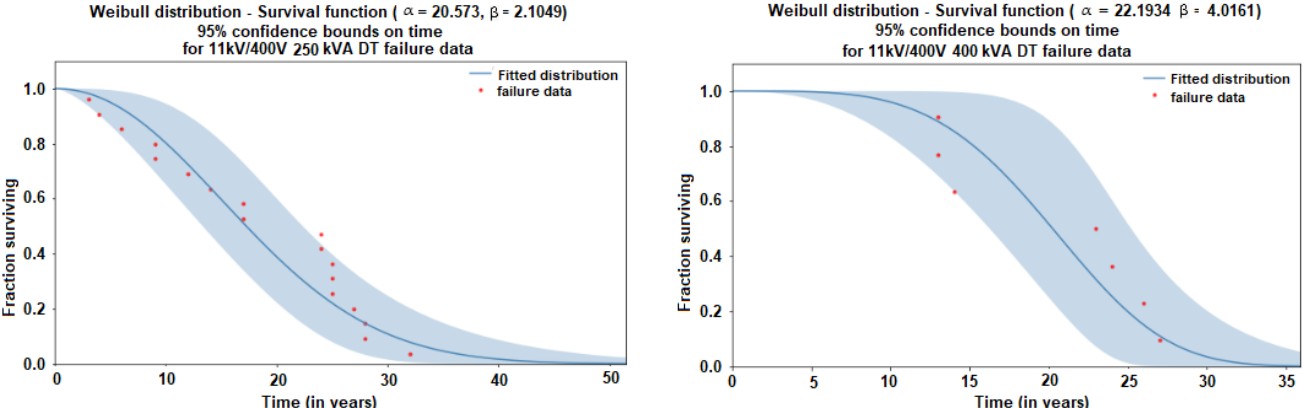

**Figure 11.** Weibull survival functions for 11 kV/0.4 V–250 kVA, 400 kVA DT failures.

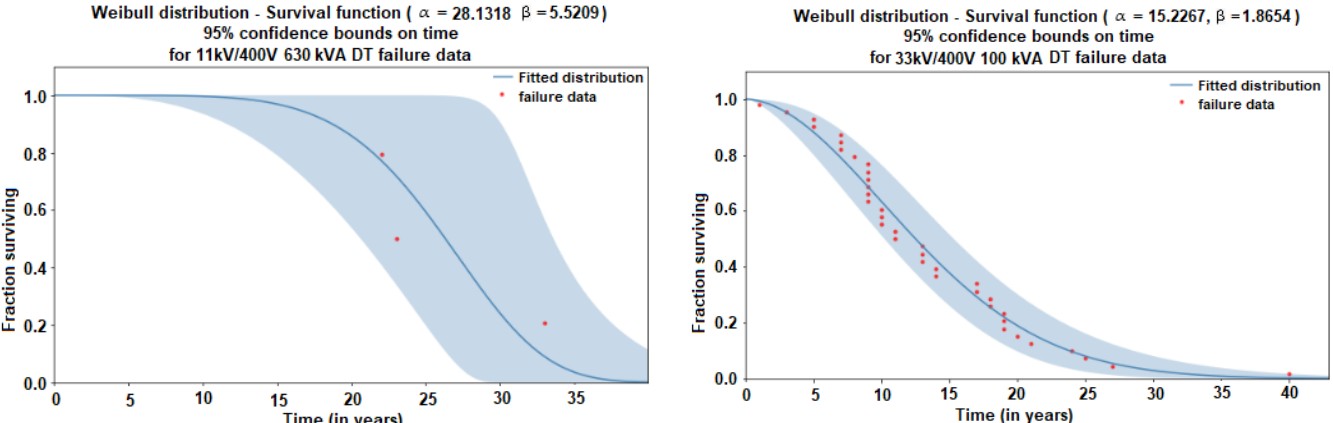

**Figure 12.** Weibull survival functions for 11 kV/0.4 V–630 kVA and 33 kV/0.4 kV–100 kVA DT failures.

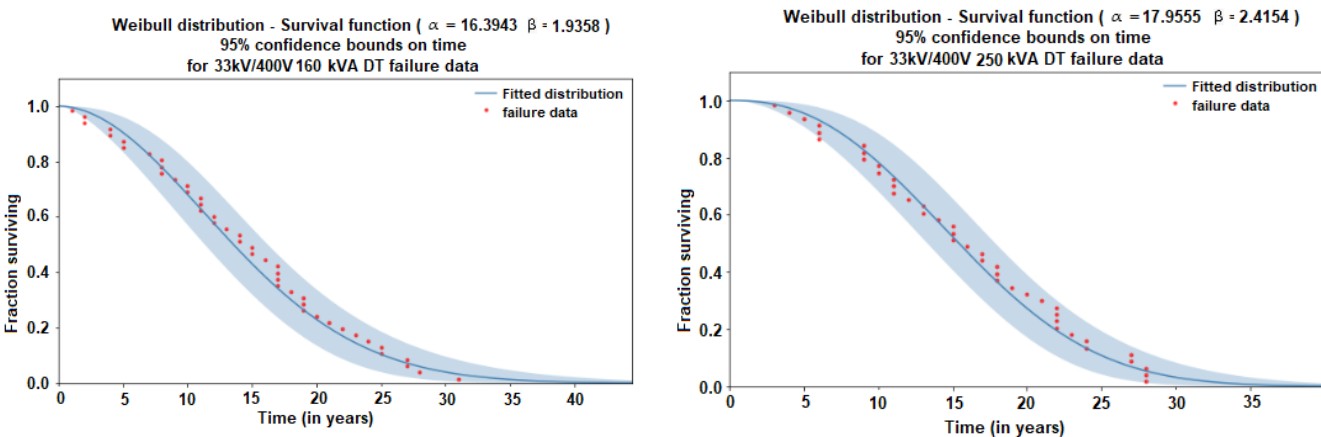

**Figure 13.** Weibull survival functions for 33 kV/0.4 kV–160 kVA, 250 kVA DT failures.

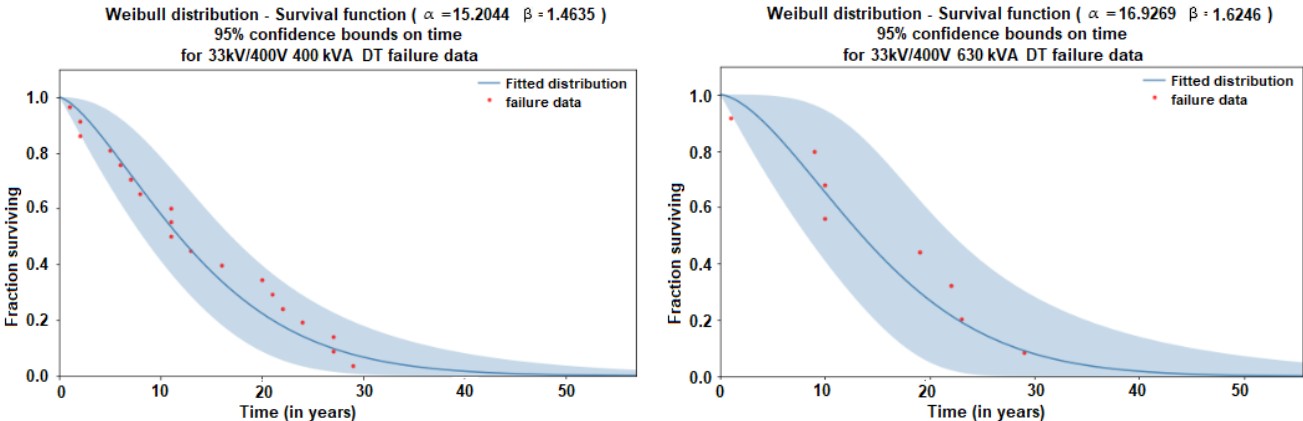

**Figure 14.** Weibull survival functions for 33 kV/0.4 kV–400 kVA, 630 kVA DT failures.

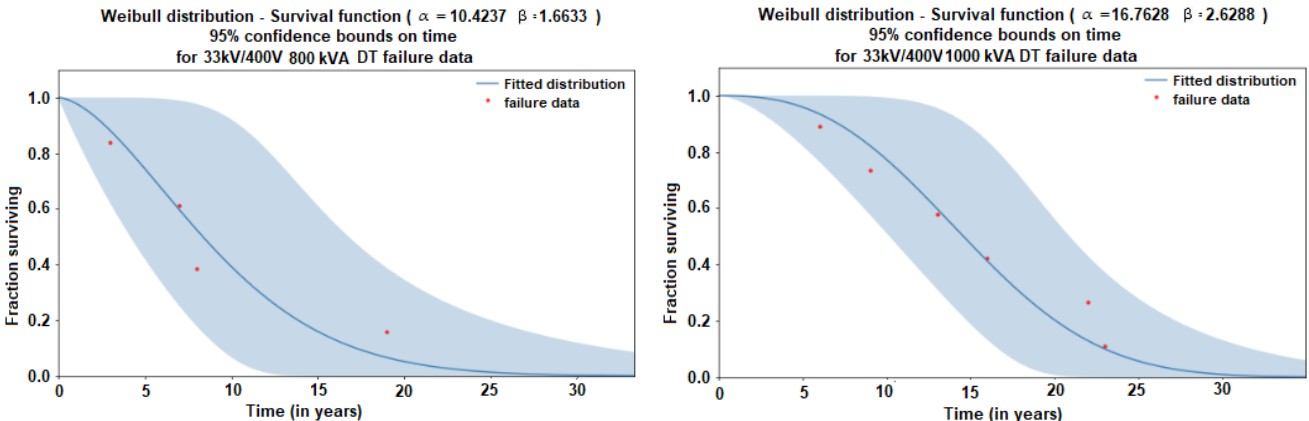

**Figure 15.** Weibull survival functions for 33 kV/0.4 kV–800 kVA, 1000 kVA DT failures.

### 3.3. Methodology and Analysis

The data were taken from the past six-year history of the transformer-category failure data from the case-study region of the case-study organization. The data were in manual format, and their digitization and digitalization were carried out for the analysis. Initially, the DT-category failure data were recorded in an Excel file. Then, using the Python program, the data from Excel were extracted and plotted in Weibull CDFs and in survival functions, using the reliability library of the Python program. The Python program calculated the

alpha and beta values for the Weibull-plot failure data. In addition, the goodness-of-fit values were generated by the same Python program.

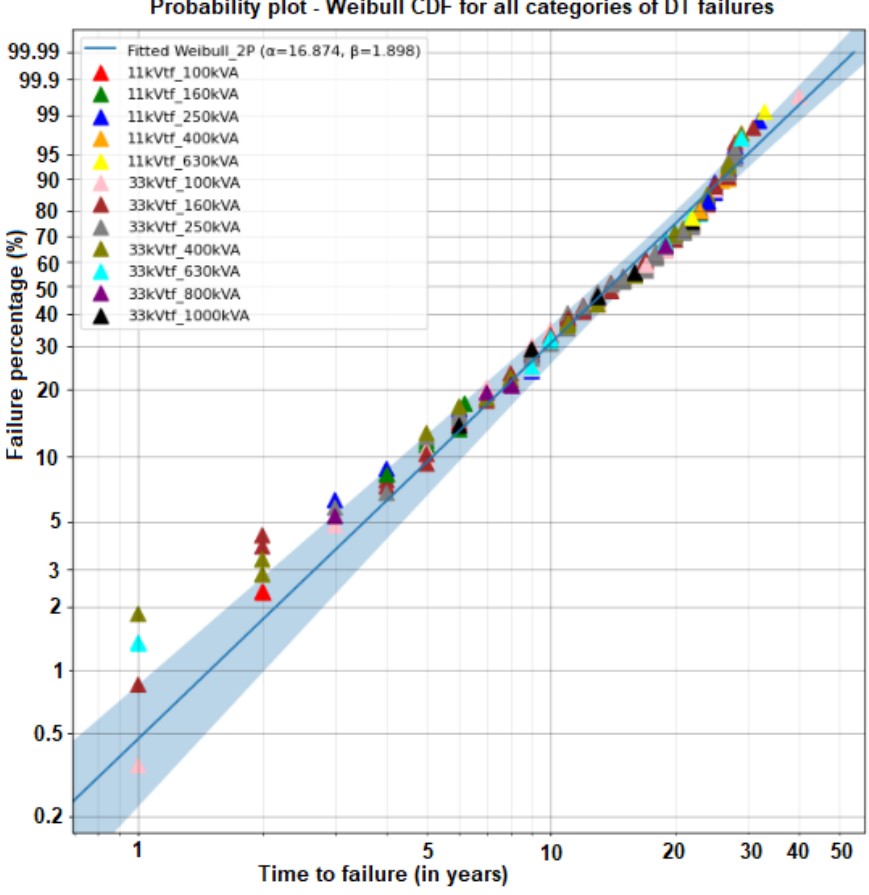

**Figure 16.** Weibull CDF for all categories of transformer failures.

Figures 4–9 above represent the Weibull CDFs of six years of failure data for 12 different categories of DTs of the selected region of the case-study organization. The data were fitted with complete data and not with incomplete (right censored) data.

In the above Weibull CDF plots, the horizontal scale provides measures of failure years for each category of DT. The vertical scale provides the cumulative percentage of the failed category of DTs. The shape factor (beta) provides an indication of the physics of the failure, and the scale (alpha) factor estimates the lifespan of DTs' characteristics. It can be either infant mortality, early wear-out, random failures, or rapid wear-out.

In addition, in the above Weibull CDF plots, the dots represent the data points that were collected as failure data for each category of DT. The line represents the theoretical CDF that was fitted to the data points based on the plotted data. The shaded area represents the uncertainty or confidence interval of the fitted Weibull distribution. It shows the range of possible values for the CDF given the uncertainty in the data and the estimation of the parameters. The thickness of the shaded area varies, depending on the amount of DT-failure data that were collected and the degree of uncertainty in the data.

Figures 4–9 above also show that, when a large amount of failure data were recorded, the uncertainty in the data were smaller and the shaded area ended up thinner in the Weibull CDFs. This is because, with more data points, the fitted Weibull distribution was able to better approximate the true underlying distribution of the DT-failure data, resulting in a more precise estimation of the parameters of the Weibull distribution. On the other hand, a small amount of DT-failure data were observed to be highly variable, and the uncertainty in the data were larger, resulting in a thicker shaded area.

Additionally, the level of confidence chosen to construct the shaded area affected how thick or thin it was. A higher level of confidence resulted in a thinner shaded area and vice versa.

The above survival functions from Figures 10–15 provide estimates of the lifespan of DTs. The survival function is plotted on the *y*-axis as a function of time on the *x*-axis. It is a decreasing function that starts at 1 and declines to 0 as time increases. The shape of the survival function is determined by the shape parameter of the Weibull distribution, and the slope of the survival function at any point is determined by the scale parameter [17,22]. These survival functions represent the probability that DTs will remain functional beyond a certain time. The survival functions are the complement of the Weibull CDFs of DTs and demonstrate the probability that the time to failure of a DT is greater than a given time t, given that the DT has not failed before time t.

In reliability and failure analysis, the survival function is more informative than the CDF, since it yields the probability of DTs still working at a certain time [21].

## 4. Results and Discussion

This section presents the results obtained with the Weibull plots implemented in this study. Comparisons of the resulting maximum likelihood estimations (MLE) presented with a truncated Newton method (TNC) optimizer for each category of distribution-transformer failures with a 96% confidence interval (CI) are shown in Table 3.

**Table 3.** Results of the Weibull analysis of DT-failure data.

| Transformer Category | No. of Failures | Parameter | Point Estimate | Standard Error | Lower CI | Upper CI |
|---|---|---|---|---|---|---|
| 11 kV to 400 V 100 kVA | 4 | Alpha | 11.4954 | 4.41194 | 5.41794 | 24.3902 |
| | | Beta | 1.37384 | 0.555899 | 0.621607 | 3.0364 |
| 11 kV to 400 V 160 kVA | 3 | Alpha | 12.7236 | 3.20768 | 7.7628 | 20.8546 |
| | | Beta | 1.48876 | 0.410304 | 0.86743 | 2.55515 |
| 11 kV to 400 V 250 kVA | 18 | Alpha | 20.573 | 2.40769 | 16.3562 | 25.8771 |
| | | Beta | 2.10485 | 0.426645 | 1.41477 | 3.13153 |
| 11 kV to 400 V 400 kVA | 7 | Alpha | 22.1934 | 2.19919 | 18.2758 | 26.9508 |
| | | Beta | 4.01613 | 1.28084 | 2.14951 | 7.50373 |
| 11 kV to 400 V 630 kVA | 3 | Alpha | 28.1318 | 3.12802 | 22.623 | 34.9819 |
| | | Beta | 5.52089 | 2.44696 | 2.31601 | 13.1607 |
| 33 kV to 400 V 100 kVA | 37 | Alpha | 15.2267 | 1.41542 | 12.6905 | 18.2696 |
| | | Beta | 1.8654 | 0.229805 | 1.46525 | 2.37485 |
| 33 kV to 400 V 160 kVA | 44 | Alpha | 16.3943 | 1.33623 | 13.9738 | 19.234 |
| | | Beta | 1.9358 | 0.240369 | 1.51763 | 2.46919 |
| 33 kV to 400 V 250 kVA | 42 | Alpha | 17.9555 | 1.20517 | 15.7422 | 20.48 |
| | | Beta | 2.41537 | 0.303852 | 1.88757 | 3.09075 |
| 33 kV to 400 V 400 kVA | 19 | Alpha | 15.2044 | 2.49815 | 11.0183 | 20.981 |
| | | Beta | 1.46352 | 0.28023 | 1.00557 | 2.13002 |
| 33 kV to 400 V 630 kVA | 8 | Alpha | 16.9269 | 3082971 | 10.8641 | 26.3731 |
| | | Beta | 1.62459 | 0.49219 | 0.897143 | 2.94189 |
| 33 kV to 400 V 800 kVA | 4 | Alpha | 10.4237 | 3.3206 | 5.5829 | 19.4618 |
| | | Beta | 1.66328 | 0.639442 | 0.782936 | 3.53352 |
| 33 kV to 400 V 1000 kVA | 6 | Alpha | 16.7628 | 2.74205 | 12.1649 | 23.0986 |
| | | Beta | 2.62885 | 0.880714 | 1.36332 | 5.06914 |

By identifying beta and alpha parameters and checking the goodness of fit, decisions can be made about the nature of the failure and its prevention. Table 4 provides information about the interpretation of the Weibull-analysis data considering the beta and alpha

parameters [27]. The shape parameter (beta) of the above Weibull distributions of the DT failures provides insight into the nature of the failure. In the above plots, a beta value less than 1 indicates that the failure rate was high at the beginning and decreased over time, which is known as infant mortality or early-life failure in the bathtub curve. In addition, a beta value equal to 1 indicates that the failure rate was constant over time, which is known as random failure. A beta value greater than 1 indicates that the failure rate was low at the beginning and increased over time, which is known as wear-out failure or late-life failure.

**Table 4.** Interpretation of Weibull CDF analysis data for DT failures [22,27].

| Beta Value | Alpha Value | Typical Failure Mode | Interpretation of Cause of Failure |
| --- | --- | --- | --- |
| >4 | Low compared with standard values for failed parts (less than 20%) | Old age, rapid wear-out (systematic, regular) | Poor machine/material design |
| Between 1 and 4 | Low compared with standard values for failed parts (less than 20%) | Early wear-out | Poor system design |
| Between 1 and 4 | Low | Early wear-out | Construction problem |
| <1 | Low | Infant mortality | Production problems, design problems, misassembled, quality control, overhaul problems |
| Between 1 and 4 | Between 1 and 4 | Less than manufacturer-recommended preventive maintenance cycle | Inadequate preventive-maintenance schedule |
| Around 1 | Much less | Random failures with definable causes | Inadequate operating procedure |

The scale parameter (alpha) of the above Weibull distributions of DT failures provides insight into the overall reliability of the system. A larger alpha value indicates a larger time to failure and higher reliability, whereas a smaller alpha value indicates a shorter time to failure and lower reliability.

Table 4 has been developed to include the above information on actual failure data of the distribution transformers of the case-study region of the case-study organization considering related alpha and beta values in the literature.

Table 5 provides the log likelihood, AIC, BIC, and AD for the goodness of fit of the Weibull model to the actual failure data set of the distribution transformer. This has been used to evaluate how well the Weibull distribution fits a given set of data. Checking the goodness of fit of the data provides valuable information. A good fit of the Weibull distribution to the data indicates that the Weibull distribution is an appropriate model for the data and that the estimated parameters had a good degree of accuracy. On the other hand, if the goodness of fit is poor, this suggests that the Weibull distribution is not a suitable model for the data and that other models should be considered.

Therefore, by identifying the beta and alpha parameters and checking the goodness of fit of the Weibull distribution to the data, a responsible officer can make better decisions about the nature of the failure and how to prevent it.

The resulting Weibull analyses of DT-failure data are to be uploaded to computerized maintenance-management systems (CMMS) for data-driven decision-making for relevant managers. With this data, corrective (reactive) maintenance work or the frequency of preventive (proactive) maintenance work can be determined for different categories of DTs. In addition, ineffective maintenance-program issues with high failure rates of DTs could easily be identified, and reliability indices (SAIDI, SAIFI, CAIDI, and ENS) of power-distribution systems could be improved with cost optimization. Also, depending on the electricity supply contracts, prioritization of the scarce resources by giving due considerations on

consumer compensations and penalty charges at the event of electricity disruptions can also be determined with this approach.

**Table 5.** Goodness of fit for the Weibull analysis of DT-failure data.

| Transformer Category | Log Likelihood | AIC | BIC | AD |
|---|---|---|---|---|
| 11 kV to 400 V 100 kVA | −13.1355 | 42.2709 | 29.0435 | 2.91429 |
| 11 kV to 400 V 160 kVA | −26.59 | 59.5801 | 12.7236 | 2.25092 |
| 11 kV to 400 V 250 kVA | −65.1467 | 135.093 | 136.074 | 1.48386 |
| 11 kV to 400 V 400 kVA | −22.1699 | 51.3398 | 48.1316 | 2.2968 |
| 11 kV to 400 V 630 kVA | −9.18554 | NA | 20.5683 | 3.76559 |
| 33 kV to 400 V 100 kVA | −19.3341 | 46.6682 | 42.2517 | 2.17347 |
| 33 kV to 400 V 160 kVA | −151.298 | 306.889 | 310.165 | 0.627865 |
| 33 kV to 400 V 250 kVA | −141.136 | 286.58 | 289.747 | 0.655464 |
| 33 kV to 400 V 400 kVA | −67.2275 | 139.205 | 140.344 | 1.05074 |
| 33 kV to 400 V 630 kVA | −28.8044 | 64.0089 | 61.7678 | 1.92899 |
| 33 kV to 400 V 800 kVA | −12.2087 | 40.4173 | 27.1899 | 2.99284 |
| 33 kV to 400 V 1000 kVA | −19.3341 | 46.6682 | 42.2517 | 2.17347 |

## 5. Conclusions

This manuscript has been developed to implement risk and reliability analysis of DTs, using failure data for maintenance planning, scheduling, and optimization by using digitalization of DT failures. The results obtained with each Weibull analysis have been compared using the MLE metrics. Digitalization of the DT data helps to determine the most suitable repair/replace strategy for DTs to improve reliability while decreasing the risk of unavailability, by making more confident data-driven decisions. Digitalization can be carried out using CMMS. From the digitalization of DT-failure data, a maintenance or reliability manager can identify maintenance strategies by leveraging historical data to improve the reliability indices (SAIDI, SAIFI, CAIDI, and ENS) of power-distribution systems without increasing costs. Furthermore, scarce resources could be managed with this approach depending on the electricity supply contracts. Hence, this methodology can be implemented at an early stage for digital transformation of power-distribution systems.

### Future Works

Future research should be carried out to evaluate the application of the other probability-distribution functions that fit the failure data of DT failures and the failures of other major components in power-distribution systems for data-driven decision-making using CMMS. This would help to adopt the optimum maintenance scheduling for power-distribution systems aiming for digital transformation.

**Author Contributions:** Writing—original draft, methodology, software, validation, and formal analysis, A.M.S.R.H.A.; writing—review, editing, and supervision, R.M.C.R. All authors have read and agreed to the published version of the manuscript.

**Funding:** This research received no external funding.

**Informed Consent Statement:** Not applicable.

**Data Availability Statement:** Not applicable.

**Conflicts of Interest:** The authors declare no conflict of interest.

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
