# Peer review of "Digitalization of Distribution Transformer Failure Probability Using Weibull Approach towards Digital Transformation of Power Distribution Systems"

_futureinternet, doi:10.3390/fi15020045_

Round 1

Reviewer 1 Report

The review report is in the attached file.

Author Response

Dear Reviewer,

Thank you very much for your valuable comments and very good improvement suggestions for our manuscript. All points that you have mentioned are equally important and responses to the points are attached in a separate file, please.

Reviewer 2 Report

The topic of the paper is interesting, although the paper must be deeply improved in terms of novelty proposed by your analysis and the description of the methodology performed and the obtained results. At the moment, too many thongs are left to the reader's imagination and guess. 

The major issues to be faced are:

-        -In the introduction you must highlight what is the novelty of your paper and the gap in literature that you are going to fill with your study;

-        -From the introduction is not clear what you propose to do in your research paper; please highlight it;

-        -In the introduction when you talk about hierarchical level of the distribution system please make also reference to:

Fusco, G.; Russo, M.; De Santis, M. Decentralized Voltage Control in Active Distribution Systems: Features and Open Issues. Energies 202114, 2563. https://doi.org/10.3390/en14092563;

S. Corsi, M. Pozzi, M. Sforna and G. Dell'Olio, "The coordinated automatic voltage control of the Italian transmission Grid-part II: control apparatuses and field performance of the consolidated hierarchical system," in IEEE Transactions on Power Systems, vol. 19, no. 4, pp. 1733-1741, Nov. 2004, doi: 10.1109/TPWRS.2004.836262.

- Also later, when you talk about Distribution Transformer please consider more reference such as:

E. R. Ronan, S. D. Sudhoff, S. F. Glover and D. L. Galloway, "A power electronic-based distribution transformer," in IEEE Transactions on Power Delivery, vol. 17, no. 2, pp. 537-543, April 2002, doi: 10.1109/61.997934.

S. Perna, G. M. Casolino and M. De Santis, "Design of a Single-Phase Two-Winding Transformer for Prototyping a Voltage Regulator," 2022 IEEE International Conference on Environment and Electrical Engineering and 2022 IEEE Industrial and Commercial Power Systems Europe (EEEIC / I&CPS Europe), 2022, pp. 1-6, doi: 10.1109/EEEIC/ICPSEurope54979.2022.9854597.

-        -Please give comments about Figure 4 to Figure 9. It is not clear what they show, besides the fact that they express Weibull CDF; what are the dots that are reported? What the line represents? What does the shadow area represent? Why in for some transformers the shadow is thicker and for other thinner?

-        -Please in the Figure 4 to 9 increase the size of the label of x and y axes, also for the legend;

-        -Why at page 8 you say “Figure 4 to Figure 15”, when you showed only up to Figure 9 since that moment?

-        -Figure 10 from Figure 15 show the “survival function” which is not clear what expresses; furthermore, there are no comments for all these Figures which is pretty strange and the rest of the paper is difficult to understand for the reader; please explain to the reader which you should imagine knows just little of it;

-        -You say that “By identifying beta and alpha parameters, checking the goodness of fit, can make decisions about the nature of the failure and its prevention”, how is that possible? Please give some methodology example that relates alpha and beta parameters to the nature characteristics of the failure; how did you make Table 2 ? it was done by comparison of your distribution parameters with experimental failures or in some other way? Please give explanation;

-        -What does Table 3 express? The values reported are large or small and how do they fit reality? Please give more comments about Table 3, which does not communicate anything to the reader;

-        -In what computer environment did you perform your analysis?

-        -How did you get the data , in what format? How did you analyze it? Please provide explanation;

Author Response

Dear Reviewer,

Thank you very much for your valuable comments and very good improvement suggestions for our manuscript. All points that you have mentioned are equally important and responses to the points are attached in a separate file.

Round 2

Reviewer 2 Report

The authors answered all the reviewer's questions.

The paper is now suitable for publication.